# HEXACONV

**Emiel Hoogeboom**[*]**, Jorn W.T. Peters**[*] **& Taco S. Cohen**
University of Amsterdam
{e.hoogeboom,j.w.t.peters,t.s.cohen}@uva.nl

**Max Welling**
University of Amsterdam & CIFAR
m.welling@uva.nl

## ABSTRACT

The effectiveness of convolutional neural networks stems in large part from their ability to exploit the translation invariance that is inherent in many learning problems. Recently, it was shown that CNNs can exploit other sources of invariance, such as rotation invariance, by using *group convolutions* instead of planar convolutions. However, for reasons of performance and ease of implementation, it has been necessary to limit the group convolution to transformations that can be applied to the filters without interpolation. Thus, for images with square pixels, only integer translations, rotations by multiples of 90 degrees, and reflections are admissible.

Whereas the square tiling provides a 4-fold rotational symmetry, a hexagonal tiling of the plane has a 6-fold rotational symmetry. In this paper we show how one can efficiently implement planar convolution and group convolution over hexagonal lattices, by re-using existing highly optimized convolution routines. We find that, due to the reduced anisotropy of hexagonal filters, planar HexaConv provides better accuracy than planar convolution with square filters, given a fixed parameter budget. Furthermore, we find that the increased degree of symmetry of the hexagonal grid increases the effectiveness of group convolutions, by allowing for more parameter sharing. We show that our method significantly outperforms conventional CNNs on the AID aerial scene classification dataset, even outperforming ImageNet pretrained models.

## 1 INTRODUCTION

For sensory perception tasks, neural networks have mostly replaced handcrafted features. Instead of defining features by hand using domain knowledge, it is now possible to learn them, resulting in improved accuracy and saving a considerable amount of work. However, successful generalization is still critically dependent on the inductive bias encoded in the network architecture, whether this bias is understood by the network architect or not.

The canonical example of a successful network architecture is the Convolutional Neural Network (CNN, ConvNet). Through convolutional weight sharing, these networks exploit the fact that a given visual pattern may appear in different locations in the image with approximately equal likelihood. Furthermore, this translation symmetry is preserved throughout the network, because a translation of the input image leads to a translation of the feature maps at each layer: convolution is translation equivariant.

Very often, the true label function (the mapping from image to label that we wish to learn) is invariant to more transformations than just translations. Rotations are an obvious example, but standard translational convolutions cannot exploit this symmetry, because they are not rotation equivariant. As it turns out, a convolution operation can be defined for almost any group of transformation — not just translations. By simply replacing convolutions with group convolutions (wherein filters are not

---

[*]Equal contribution

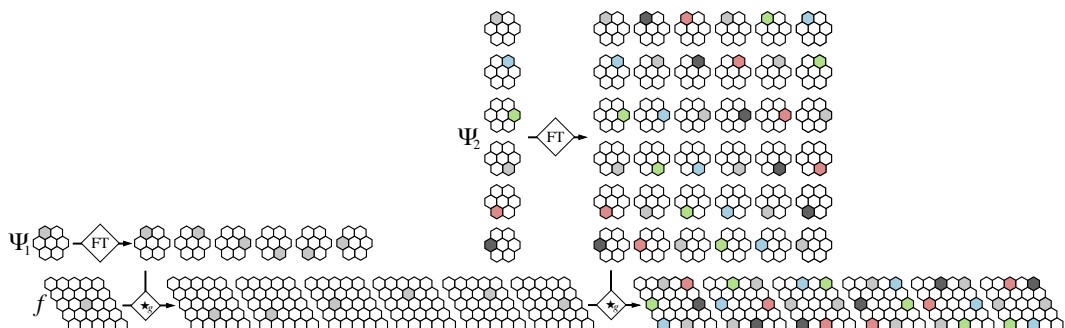

Figure 1: Hexagonal G-CNN. A $p6$ group convolution is applied to a single-channel hexagonal image $f$ and filter $\psi_1$, producing a single $p6$ output feature map $f \star_g \psi_1$ with 6 orientation channels. This feature map is then group-convolved again with a $p6$ filter $\psi_2$. The group convolution is implemented as a Filter Transformation (FT) step, followed by a planar hexagonal convolution. As shown here, the filter transform of a planar filter involves only a rotation, whereas the filter transform for a filter on the group $p6$ involves a rotation and orientation channel cycling. Note that in general, the orientation channels of $p6$ feature maps will not be rotated copies of each other, as happens to be the case in this figure.

just shifted but transformed by a larger group; see Figure 1), convolutional networks can be made equivariant to and exploit richer groups of symmetries (Cohen & Welling, 2016). Furthermore, this technique was shown to be more effective than data augmentation.

Although the general theory of such group equivariant convolutional networks (G-CNNs) is applicable to any reasonably well-behaved group of symmetries (including at least all finite, infinite discrete, and continuous compact groups), the group convolution is easiest to implement when all the transformations in the group of interest are also symmetries of the grid of pixels. For this reason, G-CNNs were initially implemented only for the discrete groups $p4$ and $p4m$ which include integer translations, rotations by multiples of 90 degrees, and, in the case of $p4m$, mirror reflections — the symmetries of a square lattice.

The main hurdle that stands in the way of a practical implementation of group convolution for a continuous group, such as the roto-translation group $SE(2)$, is the fact that it requires interpolation in order to rotate the filters. Although it is possible to use bilinear interpolation in a neural network (Jaderberg et al., 2015), it is somewhat more difficult to implement, computationally expensive, and most importantly, may lead to numerical approximation errors that can accumulate with network depth. This has led us to consider the hexagonal grid, wherein it is possible to rotate a filter by any multiple of 60 degrees, without interpolation. This allows use to define group convolutions for the groups $p6$ and $p6m$, which contain integer translations, rotations with multiples of 60 degrees, and mirroring for $p6m$.

To our surprise, we found that even for translational convolution, a hexagonal pixelation appears to have significant advantages over a square pixelation. Specifically, hexagonal pixelation is more efficient for signals that are band limited to a circular area in the Fourier plane (Petersen & Middleton, 1962), and hexagonal pixelation exhibits improved isotropic properties such as twelve-fold symmetry and six-connectivity, compared to eight-fold symmetry and four-connectivity of square pixels (Mersereau, 1979; Condat & Van De Ville, 2007). Furthermore, we found that using small, approximately round hexagonal filters with 7 parameters works better than square $3 \times 3$ filters when the number of parameters is kept the same.

As hypothesized, group convolution is also more effective on a hexagonal lattice, due to the increase in weight sharing afforded by the higher degree of rotational symmetry. Indeed, the general pattern we find is that the larger the group of symmetries being exploited, the better the accuracy: $p6$-convolution outperforms $p4$-convolution, which in turn outperforms ordinary translational convolution.

In order to use hexagonal pixelations in convolutional networks, a number of challenges must be addressed. Firstly, images sampled on a square lattice need to be resampled on a hexagonal lattice.

This is easily achieved using bilinear interpolation. Secondly, the hexagonal images must be stored in a way that is both memory efficient and allows for a fast implementation of hexagonal convolution. To this end, we review various indexing schemes for the hexagonal lattice, and show that for some of them, we can leverage highly optimized square convolution routines to perform the hexagonal convolution. Finally, we show how to efficiently implement the filter transformation step of the group convolution on a hexagonal lattice.

We evaluate our method on the CIFAR-10 benchmark and on the Aerial Image Dataset (AID) (Xia et al., 2017). Aerial images are one of the many image types where the label function is invariant to rotations: One expects that rotating an aerial image does not change the label. In situations where the number of examples is limited, data efficient learning is important. Our experiments demonstrate that group convolutions systematically improve performance. The method outperforms the baseline model pretrained on ImageNet, as well as comparable architectures with the same number of parameters. Source code of G-HexaConvs is available on Github: https://github.com/ehoogeboom/hexaconv.

The remainder of this paper is organized as follows: In Section 2 we summarize the theory of group equivariant networks. Section 3 provides an overview of different coordinate systems on the hexagonal grid, Section 4 discusses the implementation details of the hexagonal G-convolutions, in Section 5 we introduce the experiments and present results, Section 6 gives an overview of other related work after which we discuss our findings and conclude.

## 2 GROUP EQUIVARIANT CNNS

In this section we review the theory of G-CNNs, as presented by Cohen & Welling (2016). To begin, recall that normal convolutions are translation equivariant[1]. More formally, let $L_t$ denote the operator that translates a feature map $f : \mathbb{Z}^2 \to \mathbb{R}^K$ by $t \in \mathbb{Z}^2$, and let $\psi$ denote a filter. Translation equivariance is then expressed as:

$$[[L_t f] \star \psi](x) = [L_t [f \star \psi]](x). \tag{1}$$

In words: translation followed by convolution equals convolution followed by a translation. If instead we apply a rotation $r$, we obtain:

$$[[L_r f] \star \psi](x) = L_r [f \star [L_{r^{-1}} \psi]](x). \tag{2}$$

That is, the convolution of a rotated image $L_r f$ by a filter $\psi$ equals the rotation of a convolved image $f$ by a inversely rotated filter $L_{r^{-1}} \psi$. There is no way to express $[L_r f] \star \psi$ in terms of $f \star \psi$, so convolution is not rotation equivariant.

The convolution is computed by shifting a filter over an image. By changing the translation to a transformation from a larger group G, a G-convolution is obtained. Mathematically, the G-Convolution for a group $G$ and input space $X$ (e.g. the square or hexagonal lattice) is defined as:

$$[f \star_g \psi](g) = \sum_{h \in X} \sum_k f_k(h) \psi_k(g^{-1} h), \tag{3}$$

where $k$ denotes the input channel, $f_k$ and $\psi_k$ are signals defined on $X$, and $g$ is a transformation in $G$. The standard (translational) convolution operation is a special case of the G-convolution for $X = G = \mathbb{Z}^2$, the translation group. In a typical G-CNN, the input is an image, so we have $X = \mathbb{Z}^2$ for the first layer, while $G$ could be a larger group such as a group of rotations and translations. Because the feature map $f \star_g \psi$ is indexed by $g \in G$, in higher layers the feature maps and filters are functions on $G$, i.e. we have $X = G$.

One can show that the G-convolution is equivariant to transformations $u \in G$:

$$[[L_u f] \star_g \psi](g) = [L_u [f \star_g \psi]](g). \tag{4}$$

Because all layers in a G-CNN are equivariant, the symmetry is propagated through the network and can be exploited by group convolutional weight sharing in each layer.

---

[1]Technically, convolutions are exactly translation equivariant when feature maps are defined on infinite planes with zero values outside borders. In practice, CNNs are only locally translation equivariant.

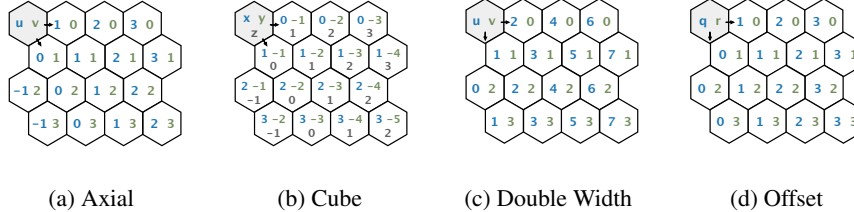

|      (a) Axial      |      (b) Cube      |   (c) Double Width   |     (d) Offset     |

Figure 2: Four candidate coordinate systems for a hexagonal grid. Notice that the cube coordinate system uses three integer indexes and both the axial and cube coordinate system may have negative indices when using a top left origin.

## 2.1 IMPLEMENTATION OF GROUP CONVOLUTIONS

Equation 3 gives a mathematical definition of group convolution, but not an algorithm. To obtain a practical implementation, we use the fact that the groups of interest can be split[2] into a group of translations ($\mathbb{Z}^2$), and a group $H$ of transformations that leaves the origin fixed (e.g. rotations and/or reflections about the origin[3]). The G-Conv can then be implemented as a two step computation: *filter transformation* ($H$) and *planar convolution* ($\mathbb{Z}^2$).

G-CNNs generally use two kinds of group convolutions: one in which the input is a planar image and the output is a feature map on the group $G$ (for the first layer), and one in which the input and output are both feature maps on $G$. We can provide a unified explanation of the filter transformation step by introducing $H_{in}$ and $H_{out}$. In the first-layer G-Conv, $H_{in} = \{e\}$ is the trivial group containing only the identity transformation, while $H_{out} = H$ is typically a group of discrete rotations (4 or 6). For the second-layer G-Conv, we have $H_{in} = H_{out} = H$.

The input for the filter transformation step is a learnable filterbank $\Psi$ of shape $C \times K \cdot |H_{in}| \times S \times S$, where $C, K, S$ denote the number of output channels, input channels, and spatial length, respectively. The output is a filterbank of shape $C \cdot |H_{out}| \times K \cdot |H_{in}| \times S \times S$, obtained by applying each $h \in H_{out}$ to each of the $C$ filters. In practice, this is implemented as an indexing operation $\Psi[I]$ using a precomputed static index array $I$.

The second step of the group convolution is a planar convolution of the input $f$ with the transformed filterbank $\Psi[I]$. In what follows, we will show how to compute a planar convolution on the hexagonal lattice (Section 3), and how to compute the indexing array $I$ used in the filter transformation step of G-HexaConv (Section 4).

## 3 HEXAGONAL COORDINATE SYSTEMS

The hexagonal grid can be indexed using several coordinate systems (see Figure 2). These systems vary with respect to three important characteristics: memory efficiency, the possibility of reusing square convolution kernels for hexagonal convolution, and the ease of applying rotations and flips.

As shown in Figure 3, some coordinate systems cannot be used to represent a rectangular image in a rectangular memory region. In order to store a rectangular image using such a coordinate system, extra memory is required for padding. Moreover, in some coordinate systems, it is not possible to use standard planar convolution routines to perform hexagonal convolutions. Specifically, in the Offset coordinate system, the shape of a hexagonal filter as represented in a rectangular memory array changes depending on whether it is centered on an even or odd row (see Figure 4).

Because no coordinate system is ideal in every way, we will define four useful ones, and discuss their merits. Figures 2, 3 and 4 should suffice to convey the big picture, so the reader may skip to Section 4 on a first reading.

---

[2]To be precise, the group $G$ is a semidirect product: $G = \mathbb{Z}^2 \rtimes H$.

[3]The group $G$ generated by compositions of translations and rotations around the origin, contains rotations around any center.

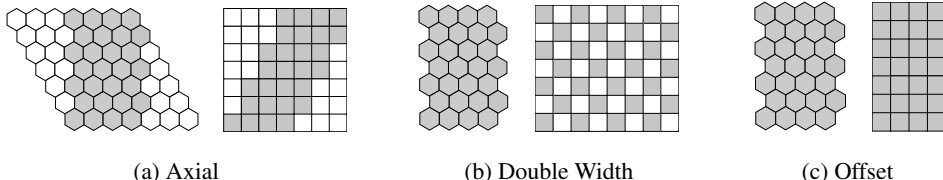

|          |                  |             |
|----------|------------------|-------------|
| (a) Axial | (b) Double Width | (c) Offset |

Figure 3: Excess space when storing hexagonal lattices, where gray cells represent non-zero values. For each coordinate system, on the left the hexagonal lattice is depicted, the grid on the right shows the 2D memory array used to store the hexagonal image.

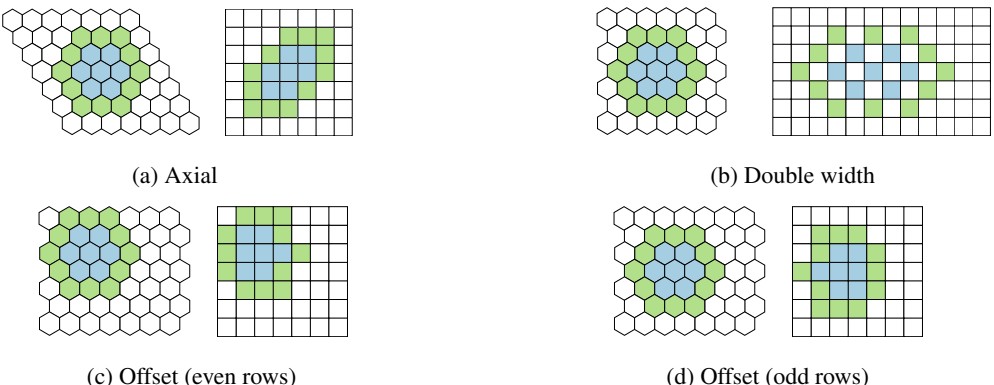

|          |          |
|----------|----------|
| (a) Axial | (b) Double width |
| (c) Offset (even rows) | (d) Offset (odd rows) |

Figure 4: Hexagonal convolution filters (left) represented in 2D memory (right) for filters of size three (blue) and five (blue and green). Standard 2D convolution using both feature map and filter stored according to the coordinate system is equivalent to convolution on the hexagonal lattice. Note that for the offset coordinate system two separate planar convolution are required — one for even and one for odd rows.

## 3.1 AXIAL

Perhaps the most natural coordinate system for the hexagonal lattice is based on the lattice structure itself. The lattice contains all points in the plane that can be obtained as an integer linear combination of two basis vectors $e_1$ and $e_2$, which are separated by an angle of 60 degrees. The Axial coordinate system simply represents the pixel centered at $ue_1 + ve_2$ by coordinates $(u, v)$ (see Figure 2a).

Both the square and hexagonal lattice are isomorphic to $\mathbb{Z}^2$. The planar convolution only relies on the additive structure of $\mathbb{Z}^2$, and so it is possible to simply apply a convolution kernel developed for rectangular images to a hexagonal image stored in a rectangular buffer using axial coordinates.

As shown in Figure 3a, a rectangular area in the hexagonal lattice corresponds to a parallelogram in memory. Thus the axial system requires additional space for padding in order to store an image, which is its main disadvantage. When representing an axial filter in memory, the corners of the array need to be zeroed out by a mask (see Figure 4a).

## 3.2 CUBE

The cube coordinate system represents a 2D hexagonal grid inside of a 3D cube (see Figure 5). Although representing grids in three dimensional structures is very memory-inefficient, the cube system is useful because rotations and reflections can be expressed in a very simple way. Furthermore, the conversion between the axial and cube systems is straightforward: $x = v$, $y = -(u + v)$, $z = u$. Hence, we only use the Cube system to apply transformations to coordinates, and use other systems for storing images.

A counter-clockwise rotation by 60 degrees can be performed by the following formula:

$$r \cdot (x, y, z) = (-z, -x, -y). \tag{5}$$

Similarly, a mirroring operation over the vertical axis through the origin is computed with:

$$m \cdot (x, y, z) = (x, z, y). \qquad (6)$$

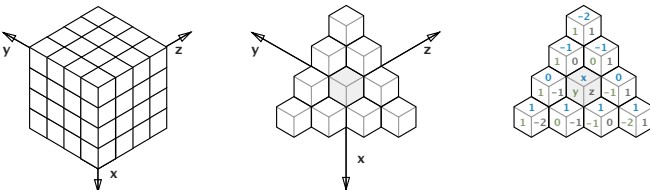

Figure 5: The cube coordinate system as a 3D structure.

### 3.3 DOUBLE WIDTH

The double width system is based on two *orthogonal* axes. Stepping to the right by 1 unit in the hexagonal lattice, the $u$-coordinate is incremented by 2 (see Figure 2c). Furthermore, odd rows are offset by one unit in the $u$ direction. Together, this leads to a checkerboard pattern (Figure 3b) that doubles the image and filter size by a factor of two.

The good thing about this scheme is that a hexagonal convolution can be implemented as a rectangular convolution applied to checkerboard arrays, with checkerboard filter masking. This works because the checkerboard sparsity pattern is preserved by the square convolution kernel: if the input and filter have this pattern, the output will too. As such, HexaConv is very easy to implement using the double width coordinate system. It is however very inefficient, so we recommend it only for use in preliminary experiments.

### 3.4 OFFSET

Like the double width system, the offset coordinate system uses two orthogonal axes. However, in the offset system, a one-unit horizontal step in the hexagonal lattice corresponds to a one-unit increment in the $u$-coordinate. Hence, rectangular images can be stored efficiently and without padding in the offset coordinate system (see Figure 3c).

The downside to offset coordinates is that hexagonal convolutions cannot be expressed as a single 2D convolution (see Figure 4c and 4d), because the shape of the filters is different for even and odd rows. Given access to a convolution kernel that supports strides, it should be possible to implement hexagonal convolution in the offset system using two convolution calls, one for the even and one for the odd row. Ideally, these two calls would write to the same output buffer (using a strided write), but unfortunately most convolution implementations do not support this feature. Hence, the result of the two convolution calls has to be copied to a single buffer using strided indexing.

We note that a custom HexaConv kernel for the offset system would remove these difficulties. Were such a kernel developed, the offset system would be ideal because of its memory efficiency.

### 4 IMPLEMENTATION

The group convolution can be factored into a filter transformation step and a hexagonal convolution step, as was mentioned in Section 2 and visualized in Figure 1. In our implementation, we chose to use the Axial coordinate system for feature maps and filters, so that the hexagonal convolution can be performed by a rectangular convolution kernel. In this section, we explain the filter transformation and masking in general, for more details see Appendix A.

The general procedure described in Section 2.1 also applies to hexagonal group convolution ($p6$ and $p6m$). In summary, a filter transformation is applied to a learnable filter bank $\Psi$ resulting in a filter bank $\Psi'$ than can be used to compute the group convolution using (multiple) planar convolutions (see the top of Figure 1 for a visual portrayal of this transformation). In practice this transformation

Table 1: CIFAR-10 performance comparison

| Model | Error | Params |
|---|---|---|
| $\mathbb{Z}^2$ | 11.50 ±0.30 | 338000 |
| $\mathbb{Z}^2$ Axial | 11.25 ±0.24 | 337000 |
| $p4$ | 10.08 ±0.23 | 337000 |
| $p6$ Axial | 9.98 ±0.32 | 336000 |
| $p4m$ | 8.96 ±0.46 | 337000 |
| $p6m$ Axial | 8.64 ±0.34 | 337000 |

Table 2: AID performance comparison

| Model | Error | Params |
|---|---|---|
| $\mathbb{Z}^2$ | 19.3 ±0.34 | 917000 |
| $\mathbb{Z}^2$ Axial | 17.8 ±0.37 | 916000 |
| $p4$ | 10.7 ±0.36 | 915000 |
| $p6$ Axial | 8.7 ±0.72 | 916000 |
| VGG (Transfer) | 9.8 ±0.50 | - |

is implemented as an indexing operation $\Psi[I]$, where $I$ is a constant indexing array based on the structure of the desired group. Hence, after computing $\Psi[I]$, the convolution routines can be applied as usual.

Although convolution with filters and feature maps laid out according to the Axial coordinate system is equivalent to convolution on the hexagonal lattice, both the filters and the feature maps contain padding (See Figure 3 and 4), since the planar convolution routines operate on rectangular arrays. As a consequence, non-zero output may be written to the padding area of both the feature maps or the filters. To address this, we explicitly perform a masking operation on feature maps and filters after every convolution operation and parameter update, to ensure that values in the padding area stay strictly equal to zero.

## 5    EXPERIMENTS

We perform experiments on the CIFAR-10 and the AID datasets. Specifically, we compare the accuracy of our G-HexaConvs ($p6$- and $p6m$-networks) to that of existing G-networks ($p4$- and $p4m$-networks) (Cohen & Welling, 2016) and standard networks ($\mathbb{Z}^2$). Moreover, the effect of utilizing an hexagonal lattice is evaluated in experiments using the HexaConv network (hexagonal lattice without group convolutions, or $\mathbb{Z}^2$ Axial). Our experiments show that the use of an hexagonal lattice improves upon the conventional square lattice, both when using normal or $p6$-convolutions.

### 5.1    CIFAR-10

CIFAR-10 is a standard benchmark that consists of 60000 images of 32 by 32 pixels and 10 different target classes. We compare the performance of several ResNet (He et al., 2016) based G-networks. Specifically, every G-ResNet consists of 3 stages, with 4 blocks per stage. Each block has two 3 by 3 convolutions, and a skip connection from the input to the output. Spatial pooling is applied to the penultimate layer, which leaves the orientation channels intact and allows the network to maintain orientation equivariance. Moreover, the number of feature maps is scaled such that all G-networks are made up of a similar number of parameters.

For hexagonal networks, the input images are resampled to the hexagonal lattice using bilinear interpolation (see Figure 6). Since the classification performance of a HexaConv network does not degrade, the quality of these interpolated images suffices.

The CIFAR-10 results are presented in Table 1, obtained by taking the average of 10 experiments with different random weight initializations. We note that the the HexaConv CNN ($\mathbb{Z}^2$ Axial) outperforms the standard CNN ($\mathbb{Z}^2$). Moreover, we find that $p4$- and $p4m$-ResNets are outperformed by our $p6$- and $p6m$-ResNets, respectively. We also find a general pattern: using groups with an increasing number of symmetries consistently improves performance.

### 5.2    AID

The Aerial Image Dataset (AID) (Xia et al., 2017) is a dataset consisting of 10000 satellite images of 400 by 400 pixels and 30 target classes. The labels of aerial images are typically invariant to rotations, i.e., one does not expect labels to change when an aerial image is rotated.

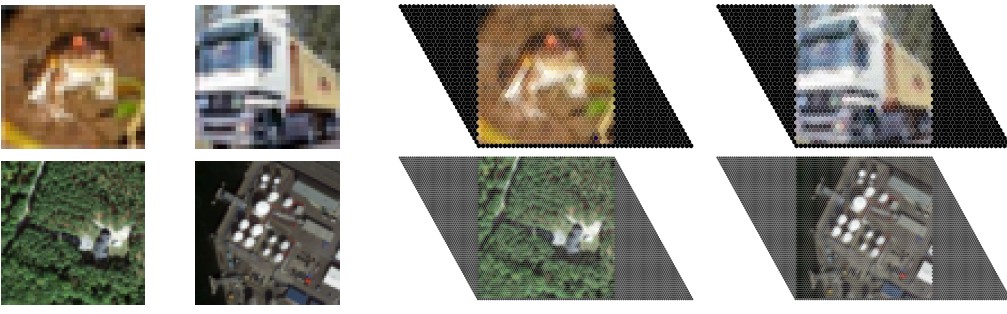

(a) Original example images        (b) Hexagonal sampled images

Figure 6: CIFAR-10 (top) and AID (bottom) examples sampled from Cartesian to hexagonal axial coordinates. Zero padding enlarges the images in axial systems.

For each experiment, we split the data set into random 80% train/20% test sets. This contrasts the 50% train/test split by Xia et al. (2017). Since the test sets are smaller, experiments are performed multiple times with randomly selected splits to obtain better estimates. All models are evaluated on identical randomly selected splits, to ensure that the comparison is fair. As a baseline, we take the best performing model from Xia et al. (2017), which uses VGG16 as a feature extractor and an SVM for classification. Because the baseline was trained on 50%/50% splits, we re-evaluate the model trained on the same 80%/20% splits.

We again test several G-networks with ResNet architectures. The first convolution layer has stride two, and the ResNets take resized 64 by 64 images as input. The networks are widened to account for the increase in the number of classes. Similar to the CIFAR-10 experiments, the networks still consist of 3 stages, but have two blocks per stage. In contrast with the CIFAR-10 networks, pooling is applied to the spatial dimensions and the orientation channels in the penultimate layer. This allows the network to become orientation invariant.

The results for the AID experiment are presented in Table 2. The error decreases from an error of 19.3% on a $\mathbb{Z}^2$-ResNet, to an impressive error of 8.6% on a $p6$-ResNet. We found that adding mirror symmetry did not meaningfully affect performance ($p4m$ 10.2% and $p6m$ 9.3% error). This suggests that mirror symmetry is not an effective inductive bias for AID. It is worth noting that the baseline model has been pretrained on ImageNet, and all our models were trained from random initialization. These results demonstrate that group convolutions can improve performance drastically, especially when symmetries in the dataset match the selected group.

## 6 RELATED WORK

The effect of changing the sampling lattice for image processing from square to hexagonal has been studied over many decades. The isoperimetry of a hexagon, and uniform connectivity of the lattice, make the hexagonal lattice a more natural method to tile the plane (Middleton & Sivaswamy, 2006). In certain applications hexagonal lattices have shown to be superior to square lattices (Petersen & Middleton, 1962; Hartman & Tanimoto, 1984).

Transformational equivariant representations have received significant research interest over the years. Methods for invariant representations in hand-crafted features include pose normalization (Lowe, 1999; Dalal & Triggs, 2005) and projections from the plane to the sphere (Kondor, 2007). Although approximate transformational invariance can be achieved through data augmentation (Van Dyk & Meng, 2001), in general much more complex neural networks are required to learn the invariances that are known to the designer a-priori. As such, in recent years, various approaches for obtaining equivariant or invariant CNNs — with respect to specific transformations of the data —were introduced.

A number of recent works propose to either rotate the filters or the feature maps followed and subsequent channel permutations to obtain equivariant (or invariant) CNNs (Cohen & Welling, 2016; Dieleman et al., 2016; Zhou et al., 2017; Li et al., 2017). Cohen & Welling (2017) describe a

general framework of equivariant networks with respect to discrete and continuous groups, based on steerable filters, that includes group convolutions as a special case. Harmonic Networks (Worrall et al., 2016) apply the theory of Steerable CNNs to obtain a CNN that is approximately equivariant with respect to continuous rotations.

Deep Symmetry Networks (Gens & Domingos, 2014) are approximately equivariant CNN that leverage sparse high dimensional feature maps to handle high dimensional symmetry groups. Marcos et al. (2016) obtain rotational equivariance by rotating filters followed by a pooling operation that maintains both the angle of the maximum magnitude and the magnitude itself, resulting in a vector field feature map. Ravanbakhsh et al. (2017) study equivariance in neural networks through symmetries in the network architecture, and propose two parameter-sharing schemes to achieve equivariance with respect to discrete group actions.

Instead of enforcing invariance at the architecture level, Spatial Transformer Networks (Jaderberg et al., 2015) explicitly spatially transform feature maps dependent on the feature map itself resulting in invariant models. Similarly, Polar Transformer Networks (Esteves et al., 2017) transform the feature maps to a log-polar representation conditional on the feature maps such that subsequent convolution correspond to group (SIM(2)) convolutions. Henriques & Vedaldi (2016) obtain invariant CNN with respect to spatial transformations by warping the input and filters by a predefined warp. Due to the dependence on global transformations of the input, these methods are limited to global symmetries of the data.

## 7 CONCLUSION

We have introduced G-HexaConv, an extension of group convolutions for hexagonal pixelations. Hexagonal grids allow 6-fold rotations without the need for interpolation. We review different coordinate systems for the hexagonal grid, and provide a description to implement hexagonal (group) convolutions. To demonstrate the effectiveness of our method, we test on an aerial scene dataset where the true label function is expected to be invariant to rotation transformations. The results reveal that the reduced anisotropy of hexagonal filters compared to square filters, improves performance. Furthermore, hexagonal group convolutions can utilize symmetry equivariance and invariance, which allows them to outperform other methods considerably.

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

## A  IMPLEMENTATION DETAILS

To understand the filter transformation step intuitively, we highly recommend studying Figure 1. Below we give a precise definition that makes explicit the relation between the mathematical model and computational practice.

Recall that in our mathematical model, filters and feature maps are considered functions $\psi : X \to \mathbb{R}^K$, where $X = \mathbb{Z}^2$ or $X = G$. In the filter transformation step, we need to compute the transformed filter $L_r\psi$ for each rotation $r \in H$ and each filter $\psi$, thus increasing the number of output channels by a factor $|H|$. The rotation operator $L_r$ is defined by the equation $[L_r\psi](h) = \psi(r^{-1}h)$. Our goal is to implement this as a single indexing operation $\Psi[I]$, where $\Psi$ is an array storing our filter, and $I$ is a precomputed array of indices. In order to compute $I$, we need to relate the elements of the group, such as $r^{-1}h$ to indices.

To simplify the exposition, we assume there is only one input channel and one output channel; $K = C = 1$. A filter $\psi$ will be stored in an $n$-dimensional array $\Psi$, where $n = 2$ if $X = \mathbb{Z}^2$ and $n = 3$ if $X = G$. An $n$-dimensional array has a set of valid indices $\mathcal{I} \subset \mathbb{Z}^n$. Thus, we can think of our array as a function that eats an index and returns a scalar, i.e. $\Psi : \mathcal{I} \to \mathbb{R}$. If in addition we have an invertible indexing function $\iota : X \to \mathcal{I}$, we can consider the array $\Psi$ as a representation of our function $\psi : X \to \mathbb{R}$, by setting $\psi(x) = \Psi[\iota(x)]$. Conversely, we can think of $\psi$ as a representation of $\Psi$, because $\Psi[i] = \psi(\iota^{-1}(i))$. This is depicted by the following diagram:

$$
\begin{array}{ccc}
\mathcal{I} & \xrightarrow{\ \Psi\ } & \mathbb{R} \\
{\scriptstyle \iota^{-1}} \Big( \Big) {\scriptstyle \iota} & \nearrow & \\
X & \swarrow {\scriptstyle \psi} &
\end{array}
\tag{7}
$$

With this setup in place, we can implement the transformation operator $L_r$ by moving back and forth between $I$ (the space of valid indices) and $X$ (where inversion and composition of elements are defined). Specifically, we can define:

$$[L_r \Psi][i] = \Psi[\iota(r^{-1}\iota^{-1}(i))] \tag{8}$$

That is, we convert our index $i$ to group element $h = \iota^{-1}(i)$. Then, we compose with $r^{-1}$ to get $r^{-1}h$, which is where we want to evaluate $\psi$. To do so, we convert $r^{-1}h$ back to an index using $\iota$, and use it to index $\Psi$.

To perform this operation in one indexing step as $\Psi[I]$, we precompute indexing array $I$:

$$I[i] = \iota(r^{-1}\iota^{-1}(i)) \tag{9}$$

Finally, we need choose a representation of our group elements $h$ that allows us to compose them easily, and choose an indexing map $\iota$. An element $h \in p6$ can be written as a rotation-translation pair $(r', t')$. The rotation component can be encoded as an integer $0, \ldots, 5$, and the translation can be encoded in the Axial coordinate frame (Section 3.1) as a pair of integers $u, v$. To compute $r^{-1}h$, we use that $r^{-1}(r', t') = (r^{-1}r', r^{-1}t')$. The rotational composition reduces to addition modulo 6 (which results in the orientation channel cycling behavior pictured in Figure 1), while $r^{-1}t'$ can be computed by converting to the Cube coordinate system and using Equation 5 (which rotates the orientation channels).

