# OpenReview forum: "HexaConv"
_ICLR.cc/2018/Conference — Accept (Poster)_

### Official Review · AnonReviewer1 · 2017-11-26
**This paper extends group equivariant convolutional networks to images with hexagonal pixelation.  While performance gains w.r.t. to the original squared lattices are not very large, the work can be inspiring for further research.**

**Rating:** 7
**Confidence:** 4

**Review:**

The paper proposes G-HexaConv, a framework extending planar and group convolutions for hexagonal lattices. Original Group-CNNs (G-CNNs) implemented on squared lattices were shown to be invariant to translations and rotations by multiples of 90 degrees. With the hexagonal lattices defined in this paper, this invariance can be extended to rotations by multiples of 60 degrees. This shows small improvements in the CIFAR-10 performances, but larger margins in an Aerial Image Dataset.

Defining hexagonal pixel configurations in convolutional networks requires both resampling input images (under squared lattices) and reformulate image indexing. All these steps are very well explained in the paper, combining mathematical rigor and clarifications.

All this makes me believe the paper is worth being accepted at ICLR conference.

Some issues that would require further discussion/clarification:
- G-HexaConv critical points are memory and computation complexity. Authors claim to have an efficient implementation but the paper lacks a proper quantitative evaluation.  Memory complexity and computational time comparison between classic CNNs and G-HexaConv should be provided.
- I encourage the authors to open the source  code for reproducibility and comparison with future transformational equivariant representations
-Also, in Fig.1, I would recommend to clarify that image ‘f’ corresponds to a 2D view of a hexagonal image pixelation.  My first impression was a rectangular pixelation seen from a perspective view.

---

> ### Author Response · Authors · 2018-01-01
> **reply to reviewer 1**
>
> Dear reviewer,
>
> Thank you for your support and comments.
>
> On memory & computation complexity:
> In our method, the memory and computational complexity scale as in the framework introduced by Cohen and Welling. Say n is the number of elements in a group (e.g. 6 rotations), and say we wish to keep the number of parameters fixed relative to a planar CNN. Then memory scales with sqrt(n) (e.g. ~2.5), and computational complexity scales with n. This is exactly the same cost as simply increasing the number of channels by ~2.5, which one would normally do only when the dataset was much larger.
>
> On open source:
> To facilitate the development of further research, in areas such as G-HexaConvs on other coordinate systems, we will release our source code on github. This also addresses the second point that the reviewer raises.
>
> On Figure 1:
> To improve the clarity of Fig. 1, we modified the borders and size. In addition, the caption also describes what the image f is. We hope that this addresses the reviewer’s concerns regarding the figure.

---

### Official Review · AnonReviewer2 · 2017-11-26
**Interesting general approach, not really convinced by the practical use**

**Rating:** 7
**Confidence:** 4

**Review:**


The authors took my comments nicely into account in their revision, and their answers are convincing. I increase my rating from 5 to 7. The authors could also integrate their discussion about their results on CIFAR in the paper, I think it would help readers understand better the advantage of the contribution.

----

This paper is based on the theory of group equivariant CNNs (G-CNNs), proposed by Cohen and Welling ICML'16.

Regular convolutions are translation-equivariant, meaning that if an image is translated, its convolution by any filter is also translated. They are however not rotation-invariant for example.  G-CNN introduces G-convolutions, which are equivariant to a given transformation group G.

This paper proposes an efficient implementation of G-convolutions for 6-fold rotations (rotations of multiple of 60 degrees), using a hexagonal lattice. The approach is evaluated on CIFAR-10 and AID, a dataset of aerial scene classification. The approach outperforms G-convolutions implemented on a squared lattice, which allows only 4-fold rotations on AID by a short margin. On CIFAR-10, the difference does not seem significative (according to Tables 1 and 2).
I guess this can be explained by the fact that rotation equivariance makes sense for aerial images, where the scene is mostly fronto-parallel, but less for CIFAR (especially in the upper layers), which exhibits 3D objects.

I like the general approach of explicitly putting desired equivariance in the convolutional networks. Using a hexagonal lattice is elegant, even if it is not new in computer vision (as written in the paper). However, as the transformation group is limited to rotations, this is interesting in practice mostly for fronto-parallel scenes, as the experiences seem to show. It is not clear how the method can be extended to other groups than 2D rotations.

Moreover, I feel like the paper sometimes tries to mask the fact that the proposed method is limited to rotations. It is admittedly clearly stated in the abstract and introduction, but much less in the rest of the paper.

The second paragraph of Section 5.1 is difficult to keep in a paper. It says that "From a qualitative inspection of these hexagonal interpolations we conclude that no information is lost during the sampling procedure."  "No information is lost" is a strong statement from a qualitative inspection, especially of a hexagonal image.  This statement should probably be removed. One way to evaluate the information lost could be to iterate interpolation between hexagonal and squared lattices to see if the image starts degrading at some point.

---

> ### Author Response · Authors · 2018-01-01
> **reply to reviewer 2**
>
> Dear reviewer,
>
> Thank you for your review.
>
> On performance of G-HexaConvs:
> In the experiments we show that performance consistently improves with increasing degrees of symmetry. We understand the concern of the reviewer that these differences are small for the CIFAR dataset. The results of the experiments were collected over 10 different runs with random parameter initializations. The experiments section of the paper has been updated to emphasize that the values are obtained by taking the average of 10 runs. To show statistical significance of 6-fold rotational symmetries over 4-fold rotational symmetries, we have done a significance test on our data. We test p4 and p4m versus p6 and p6m (our method) in a pairwise t-test, and find it passes with p=0.036.
>
> Also, it should not be undervalued that our method outperforms a transfer learning approach on AID, that has been pretrained on ImageNet. Our method reduces classification error by 2% compared to networks that leverage only 4-fold symmetries. And our methods improve the error of conventional network by more than 11%.
>
> On extensions to other groups than 2D rotations:
> The reviewer is right to observe that in fronto-parallel scenes, this method can leverage global symmetries in the picture. Nonetheless, our experiments on CIFAR-10 show that although the margin of the benefits is smaller, our method can leverage local symmetries on a smaller scale and improve performance. These findings agree with earlier experiments by Cohen and Welling who used only 4-fold symmetries.
>
> On masking limitations to the group of rotations:
> It is not our intention to mask in any way that our method is limited to mirror and rotation transformations. Note that although the mathematical framework introduced by Cohen and Welling can be used for any group, in some cases, such the case of 6-fold rotational symmetry, the concrete implementation is far from trivial. Our paper is focused on the various data structures and indexing schemes that are required for an efficient implementation of hexagonal G-convolutions. If the reviewer is not entirely satisfied after the updates we made to the paper, perhaps the reviewer can help us by pointing to specific locations that could be improved in this respect.
>
> On information loss conclusion by qualitative inspection:
> We completely agree with the reviewer this is not a precise claim. None of the conclusions on our paper depend on this claim. Moreover, classification performance does not degrade when using hex-images. The paragraph is rephrased in the updated paper.

---

### Official Review · AnonReviewer3 · 2017-11-27
**A good submission that shows how to practically implement G-convolutional layers for DNNs on hexagonal lattices and the benefits of doing so.**

**Rating:** 7
**Confidence:** 4

**Review:**

The paper presents an approach to efficiently implement planar and group convolutions over hexagonal lattices to leverage better accuracy of these operations due to reduced anisotropy. They show that convolutional neural networks thus built lead to better performance - reduced inductive bias - for the same parameter budget.

G-CNNs were introduced by Cohen and Welling in ICML, 2016. They proposed DNN layers that implemented equivariance to symmetry groups. They showed that group equivariant networks can lead to more effective weight sharing and hence more efficient networks as evinced by better performance on CIFAR10 & CIFAR10+ for the same parameter budget. This paper shows G-equivariance implemented on hexagonal lattices can lead to even more efficient networks.

The benefits of using hexagonal lattices over rectangular lattices is well known in the signal processing as well as in computer vision. For example, see

Golay M. Hexagonal parallel pattern transformation. IEEE Transactions on Computers 1969. 18(8): p. 733-740.

Staunton R. The design of hexagonal sampling structures for image digitization and their use with local operators. Image and Vision Computing 1989. 7(3): p. 162-166.

L. Middleton and J. Sivaswamy, Hexagonal Image Processing, Springer Verlag, London, 2005

The originality of the paper lies in the practical and efficient implementation of G-Conv layers. Group-equivariant DNNs could lead to more robust, efficient and (arguably) better performing neural networks.

Pros

- A good paper that systematically pushes the state of the art towards the design of invariant, efficient and better performing  DNNs with G-equivariant representations.

- It leverages upon the existing theory in a variety of areas - signal & image processing and machine learning, to design better DNNs.

 - Experimental evaluation suffices for a proof of concept validation of the presented ideas.


Cons

- The authors should relate the paper better to existing works in the signal processing and vision literature.

- The results are on simple benchmarks like CIFAR-10. It is likely but not immediately apparent if the benefits scale to more complex problems.

- Clarity could be improved in a few places

: Since * is used for a standard convolution operator, it might be useful to use *_g as a G-convolution operator.

: Strictly speaking, for translation equivariance, the shift should be cyclic etc.

: Spelling mistakes - authors should run a spellchecker.

---

> ### Author Response · Authors · 2018-01-01
> **reply to reviewer 3**
>
> Dear reviewer,
>
> We thank you for your comments and suggestions.
>
> On hexagonal literature:
> We agree that it is important to recognize existing work on hexagonal signal processing, and have added references by Petersen, Hartman and Middleton in the updated paper.
>
> On benefits to scaling:
> We agree that CIFAR-10 is a relatively simple benchmark. In our experiments we do show that an identical network architecture where conventional conv layers are replaced by hexagonal g-conv layers, results in consistent improvements on two distinct datasets. Furthermore, we plan to release our codebase which can help further research to scale these methods to larger problems.
>
> On the group convolution operator:
> We think it is a very good suggestion to change group convolution operators from * to *_g, to clarify what type of convolution is used. We changed the relevant operators in the updated paper.
>
> On exact translation equivariance in CNNs:
> We agree with the reviewer that in order for equivariance to hold exactly, either:
> shifts should be cyclic, or
> One should use “valid”-mode convolutions and consider the input image as a function defined on all of Z^2, where values outside of the original image are zero.
> In practice, we use “same” convolution instead of “valid” convolution, because the latter would increase the size of the feature maps with each layer. Thus, a typical convolutional network is not exactly translation equivariant. We have added a footnote that addresses this detail.
>
> On spellchecking:
> We have run a spellchecker and fixed the spelling mistakes.

---

### Public Comment · (anonymous) · 2017-12-13
**Comparison with other CNNs that use hex. shape kernels.**

As the reviewers suggested, hex. shape kernels have been used for vision tasks for a long decade.

In the following paper, hex. shape kernels were used to train CNNs for image classification and detection, using Cifar 10/100 and Imagenet datasets:

Z. Sun, M. Ozay, T. Okatani, Design of Kernels in Convolutional Neural Networks for Image Classification, ECCV 2016.

There are also works on adaptive convolution, which employ variations of adaptive shape convolution operations, for instance;

S. Niklaus, L. Mai, F. Liu, Video Frame Interpolation via Adaptive Convolution, CVPR 2017.

S. Niklaus, L. Mai, F. Liu, Video Frame Interpolation via Adaptive Separable Convolution, ICCV 2017.

J. Dai, H. Qi, Y. Xiong, Y. Li, G. Zhang, H. Hu, and Y. Wei, Deformable Convolutional Networks, ICCV 2017.

- What is the novelty of HexaConv compared to these previous works (i.e. Hex. kernels and adaptive/deformable convolution)? A detailed comparison is required in order to get the superiority of the proposed HexaConv.

-  The performance of HexaConv is less than the perf. of these sota methods. Could you please provide a more detailed analysis, esp. using HexaConv on larger datasets of natural images, e.g. Imagenet?

---

> ### Author Response · Authors · 2018-01-01
> **reply to public comment**
>
> Dear commenter,
>
> Thank you for your interest in our paper.
>
> Although hexagonal grids have been used in signal processing for some time, our work is focused on the implementation of the group convolution for 6-fold rotational groups p6 and p6m. Thus, unlike other methods, our approach is able to exploit the symmetries of the hexagonal grid to improve statistical efficiency by parameter sharing. We have shown that our method convincingly beats a solid baseline on CIFAR, and outperforms the transfer learning baseline on AID. Our method is not related to adaptive, deformable, or separable convolution in either approach or intent.

---

### Decision · Program_Chairs · 2018-01-29
**ICLR 2018 Conference Acceptance Decision**

**Decision:**

Accept (Poster)

**Comment:**

This paper implements Group convolutions on inputs defined over hexagonal lattices instead of square lattices, using the roto-translation group. The internal symmetries of the hexagonal grid allow for a larger discrete rotation group than when using square pixels, leading to improved performance on CIFAR and aerial datasets.

The paper is well-written and the reviewers were positive about its results. That said, the AC wonders what is the main contribution of this work relative to existing related works (such as Group Equivarant CNNS, Cohen & Welling'16, or steerable CNNs, Cohen & Welling'17). While it is true that extending GCNNs to hexagonal lattices is a non-trivial implementation task, the contribution lacks significance in the mathematical/learning fronts, which are perhaps the ones ICLR audience will care more about. Besides, the numerical results, while improved versus their square lattice counterparts, are not a major improvement over the state-of-the-art.

In summary, the AC believes this is a borderline paper. The unanimous favorable reviews tilt the decision towards acceptance.